# Recovering Sexuality after Childbirth. What Strategies Do Women Adopt? A Qualitative Study

**DOI:** 10.3390/ijerph19020950

**Published:** 2022-01-15

**Authors:** Esther Delgado-Pérez, Isabel Rodríguez-Costa, Fernando Vergara-Pérez, María Blanco-Morales, María Torres-Lacomba

**Affiliations:** 1Physiotherapy Department, Faculty of Sport Sciences, Universidad Europea de Madrid, Villaviciosa de Odón, 28670 Madrid, Spain; esther.delgado@universidadeuropea.es (E.D.-P.); maria.blanco@universidadeuropea.es (M.B.-M.); 2Humanization in the Intervention of Physiotherapy for the Integral Attention to the People (HIPATIA) Research Group, Physiotherapy Department, Faculty of Medicine and Health Sciences, University of Alcalá, Alcalá de Henares, 28801 Madrid, Spain; 3Physiotherapy in Women’s Health (FPSM) Research Group, Physiotherapy Department, Faculty of Medicine and Health Sciences, University of Alcalá, Alcalá de Henares, 28801 Madrid, Spain; fernando.vergara@uah.es (F.V.-P.); maria.torres@uah.es (M.T.-L.)

**Keywords:** strategies, sexuality, intimacy, postpartum, emotional management, motherhood

## Abstract

This study aimed to determine the strategies used by women to adapt to the changes that affect the first sexual relations after childbirth. A qualitative study with a phenomenological approach used three data collection techniques (in-depth interviews, discussion groups, and online forums). Thirty-six women in the first six months postpartum participated in the study, from physiotherapy centers with maternal child specialties in several locations in Spain. Women with different types of delivery, presence or absence of perineal trauma during delivery, previous deliveries, and different types of breastfeeding were included. Among the strategies, closeness support and understanding were the ones that women used to adjust to the new situation, in order to improve the couple’s relationship, intimacy, and cope with the significant changes that appear in the first six months postpartum. Changes and adaptations in sexual practices become a tool for coping with a new sexuality, especially if it is affected by the presence of pain or discomfort associated with physical changes. Personal time facilitates emotional management and improvement of emotional changes related to the demands of motherhood. Accepting the changes that motherhood brings is critical to dealing with the new situation. Strategies used by postpartum women focus on acceptance, self-care, partner, couple time, personal time, and adapting encounters. The findings of this study are of interest to health professionals as they provide insight into how women cope with the changes that appear in the first six months postpartum. In this way, the findings will be able to transmit to couples the alternatives they can adopt before the resumption of sexual relations to improve satisfaction both as a couple and in terms of sexuality after childbirth.

## 1. Introduction

Motherhood implies adapting to new situations, including sexual relations [1]. Sexuality should be understood as a central aspect of the human being that is present throughout life [2] and is perceived differently at each stage. During the postpartum period, women face multiple physical, emotional, and relational changes that inevitably affect their sexuality [3]. The postpartum period is described, from a medical point of view, as the time needed after childbirth for the reproductive organs to return to their pre-pregnancy state, being approximately six weeks in length [4].

The estimated time to resume sexual relations after childbirth varies between 6 and 8 weeks, but only 32% of those surveyed took up sexual relations again in this period [5,6]. Women may tend to delay sexual relations because they are not yet ready, due to the changes they are experiencing [7,8,9]. Although most women tend to return to sexual relations between the third and sixth month postpartum, some authors suggest that many women experience at least one sexual problem [10,11,12,13]. 

The challenges faced by women during this period are largely due to the demanding nature of postpartum. After childbirth, women present physical factors such as the presence of perineal trauma, with dyspareunia being the factor that affects sexual functionality and satisfaction [9,10,14,15,16,17,18,19,20,21]. Common emotional factors at this stage also influence the experience of sexuality, including depression, the responsibility of assuming new roles, fears that accompany motherhood, and self-image distortion [7,19]. In addition, the sexuality of postpartum women is impaired by fatigue, extreme tiredness, lack of time for oneself or intimacy with the partner [22,23].

Some studies show that these factors (physical, emotional, and vital) are responsible for women showing reduced sexual interest or desire, lack of lubrication, arousal, and changes in sensitivity that make pleasure or orgasm more difficult, which implies less sexual satisfaction for the couple [9,14,19,24,25]. Sexual activity progressively resumes between the first 6 weeks and 6–12 months after childbirth, but previous levels of sexual function and frequency will not tend to normalize until six months after childbirth [26].

Most studies focus on identifying the changes that occur during this period that are responsible for sexual health changes [8,15,27,28,29], although few studies analyze women’s coping strategies or consider women’s own perspective under a biopsychosocial framework. The study by O´Malley et al. exposed strategies at 12 months postpartum before the restart of coital relationships, the management of physical changes, as well as showing the strategies for overcoming psychological changes [1]. In addition, Bender et al. described women’s needs for intimacy, communication, and closeness in dealing with sexual experience at 6 and 12 months postpartum [7]. The study by Priddis et al. was performed on women who had suffered perineal trauma, described some of the strategies women use to cope with the new reality between 7 weeks and 12 years postpartum [20]. However, none of the articles focus on the strategies that women adopt in sexual relations during the first six months postpartum, despite this period being the time when women have their first sexual encounter after childbirth [5,6,10,11,12,13]. 

The current study tried to determine the strategies used by Spanish women to adapt to the changes that affect the first sexual encounters in the first six months after childbirth. 

## 2. Materials and Methods

### 2.1. Study Design

The study used a qualitative, phenomenological approach with a descriptive perspective [30,31] to observe the phenomenon. Through this study we sought to contextualize and better understand the phenomena and to gain insight into the subjective experience of people. The research was carried out between June 2017 and May 2019. Five researchers conducted the study, which included two physiotherapists (EDP and MTL) with expertise in women’s health and three researchers (IRC, FVP, and MBM) with experience in qualitative research. The researchers whose specialized work is developed in the field of maternity focused on postpartum care, have been observing the sexuality problems that women face in this period, in addition to the poor professional attention they receive. This way, the need arises to deeply evaluate the way many women adapt or adjust to the changes that occur in the 6 months and directly affect sexuality.

### 2.2. Participants

The women were recruited in the context of private physiotherapy centers located in three different locations in Spain, two centers in Madrid, another in Toledo, and one in Talavera de la Reina. The selected centers had a specific unit in maternity and childcare, where the collaborating researchers carried out the recruitment of participants. The women had to be of legal age, and physically and mentally capable of understanding and participating in the study and understanding the Spanish language. Women who had given birth in the last 6 months were included, and they were not excluded based on how they had given birth, presence or not of perineal trauma during delivery, previous deliveries, or type of breastfeeding. All women with systemic (such as diabetes, multiple sclerosis, lupus erythematosus) or neurological diseases and/or cognitive problems were excluded, along with those mothers who, during childbirth, experienced problems requiring hospitalization or loss of the baby due to perinatal death. The sample was selected by theoretical sampling [32] following a convenience criterion by physiotherapists from private physiotherapy centers in each locality. All participants completed the study. 

### 2.3. Data Collection

A researcher (EDP) carried out the data collection through different methods: (1) two conversation techniques: in-depth interviews [33] and discussion groups [34]; and (2) a technique based on observation, through an online forum [35]. For data development, the researcher (EDP) minimally guided each interview, discussion group, and forum, using a semi-structured interview guide (Table 1). These topics were agreed upon by an expert group of four physiotherapists: two qualitative research experts and two experts in maternity.

The place and the day of the interviews or discussion groups were agreed upon by the participants, being chosen preferably in a quiet room or the most intimate place where the participant feels comfortable, such as the participant’s home, and, in the case of the discussion groups, in the collaborating physiotherapy centers. Concerning the online forum, interactions took place in a private room using the Facebook interface. Facebook room was called “Sex After Childbirth” and was active between March and May 2018, where the participating women (who had been invited by mail through a snowball sampling) presented their experiences over a full week on the topics, as shown in Table 1. 

### 2.4. Data Analysis 

All interviews were recorded and transcribed entirely by a member of the research team (EDP). For the coding of the fragments, the qualitative analysis software MAXQDA in its 2018 version was used, which facilitated the interactive process on which the theorization has been based [36]. After an in-depth reading, three members of the research team (EDP, IRC, and MTL) carried out an open, axial, and selective coding [37]. The fragments were analyzed in Spanish and subsequently translated to English. All reports collected (via in-deep interview, discussion group, and online forum) were treated equally. Subsequently, the research team worked together to regroup and agree on the topics through the creation of a code book [38], with the aim of generating a conceptual framework that would explain the phenomenon. Through the discussion group with mothers, and thanks to the contribution of participant DGM31 (discussion group—DG, mother—M, and participant number—31), the experiences reported by the women did not generate new concepts, thereby reaching theoretical saturation. All data were processed following the validation process based on [36] regulated criteria to establish scientific rigor: credibility, transferability, dependence, and confirmability [39,40]. We were able to guarantee: the triangulation of methods, researchers, and locations; the reflection of the research team; the recording of the phenomenon; being as precise as possible; and the description of the methodology developed. Finally, the Standards for Reporting Qualitative Research (SRQR) guidelines [41] and COREQ checklist [42] were used to increase scientific rigor.

## 3. Results

The sample consisted of 36 Spanish white women who followed the entire study without a drop out. The sociodemographic data, as well as the gynecological and obstetric data, are shown in Table 2 and Table 3, respectively. In relation to participation, 14 women formed the sample of these discussion groups that were conducted in Madrid (5 women), Toledo (4 women), and Talavera de la Reina (5 women); 12 women were interviewed individually; and 10 women participated in the online forum. The mean age of the 36 women was 34 years (within the range of 27 to 39 years), and the mean postpartum period ranged from 2 weeks to 4 months.

Faced with the physical, emotional, and vital changes characteristic of the postpartum period, women look for alternatives or strategies to improve their sexual satisfaction during the first six months after childbirth and manage the new situation. Through the process of analyzing the collected material, 17 codes were identified and grouped into 3 categories. One of the categories was divided into two subcategories, formed by 11 of the 17 codes (Table 4). A total of 334 units of meaning related to the strategies used by women to cover physical, emotional, and vital changes affecting sexuality were obtained. 

### 3.1. Strategies to Address Physical Changes Affecting Sexuality

This section explores the strategies women use to enhance their sexuality in the face of the physical changes they experience after childbirth. Using lubricants, adopting new postures, and avoiding postures that may lead to pain or discomfort, and even changing the pace or depth are the strategies most widely used by these women to manage physical changes affecting sexual relationships and to facilitate sexual encounters.


*“So, in the end we never… we never get to have sexual intercourse as you said before, so maybe we do more sex games and other kinds of things, but it still hurts me.”*

*(IM4, in-depth interview with mothers).*



*“Well, that’s in some positions, when I’m on top, for example, there are many times when it hurts me, just as I don’t know, a pain, I don’t know, here, and then I change position, or maybe in some positions like “on all fours”, I also feel pain, I don’t know, so that’s it, I do feel pain in those two positions.”*

*(IM5, in-depth interview with mothers).*


These strategies arise to manage the presence of physical alterations, such as pain (Table 4), and the fear of suffering it, or the lack of lubrication. Women related this pain to sexual practices that were associated with penetration, and it was present, for many of them, from the beginning of the relationships. They described this pain as sharp, with a feeling of tightness and often as a stinging sensation, as well as feeling it with great intensity. Other women did not talk about pain, but as a constant discomfort that does not allow them to feel pleasure in sexual relations. Women considered the presence of the scar (episiotomy or tear), lack of lubrication, friction, or depth of penetration as possible causes. On many occasions, women use sexual relations to test the condition of their genital area and take measures to improve.


*“No, look, I think it has been a challenge… because previously eh… when… when I started sexual relations the same thing was happening to me, it hurt quite a lot… and as I continued having relationships I hurt less and less. So, it has been more of a challenge than a fear, a saying, well, it’s like I’ve had to re-educate my entire genital system to sex again […]”*

*(IM7, in-depth interview with mothers).*



*“My first motivation was almost to “test” myself to see if I had a lot of pain- tension in the area of the episiotomy or the vaginal introitus.”*

*(OF17, online forum to mothers).*


### 3.2. Strategies for Dealing with Emotional Changes Affecting Sexuality

This section sets out the strategies that women describe that they have used to cope with perceived emotional changes through two subcategories, which focus on coping with body image and emotion management, as shown in Table 4.

Women use strategies focused on self-care and exercise to feel better in response to physical changes and the subsequent impact on body image (Table 4). Physical changes are generally perceived throughout the body, especially in the breast and genital area, affecting their self-esteem and creating insecurities that extend to their partner.


*“And not only about doing sport, but about getting up and telling you… my boy J* breaks, because I get up and I paint my eye line and, in the hospital, I paint myself because if not… I mean, it’s an idea I have, as I look bad, because I sink. Well, I get up and paint my eye line and my mascara, just like I put my clothes on, because if I don’t look bad, and if you look deadly, then you’ll see… it’s true!”*

*(DGM36, discussion group to mothers).*


They describe with significant emphasis the need to feel themselves prior to childbirth and to feel attractive and desired by their partners as fundamental to improve their perception and self-esteem.


*“Eh… well, the other day I already had the need to put on a bra of mine that wasn’t a nursing bra, because it’s as if you’re not looking good in a horrible nursing bra, with the discs, so it doesn’t help. So, I’ve already decided that I’m going to wear my bra when I feel like it (laughs) […] But it’s very important because it’s about starting to look like you used to look and to look good”.*

*(IM8, in-depth interview with mothers).*



*“Yes, what happens is that he also raises my self-esteem a lot in that aspect, so, well, to feel that he likes me makes me feel better.”*

*(IM8, in-depth interview with mothers).*


On the other hand, the women described strategies for emotional management, such as: attendance and participation in meetings with other postpartum women; the support provided by their partner through good communication; the positive acceptance of the changes in the postpartum period; the enjoyment of an individual time in which to enjoy themselves and take care of themselves through walks, exercise, or meeting friends; and, finally, the performance of physical activity.

The puerperium is a time of great emotional intensity due to the emotional fluctuations that are related to the state of mind that is governed by emotional ups and downs, the duality of motherhood, and even by the presence of puerperal sadness. In addition to these changes that they experience, it is a time when women feel a great insecurity related to the care of the baby, the stress suffered by the couple, the pressure exerted by the environment and society, and the demands that the mother herself imposes.

Meetings with other women give confidence and serenity to listen and share opinions and experiences with other women who are living the same stage. At these meetings, they can talk about baby care, their emotions, and situations they are dealing with.


*“I’m also helped by the opinions of other mums who have been through the same thing, it’s a new world for me and unknown.”*

*(OFM20, online forum to mothers).*



*“[…] yes, you get together with mothers, and you can talk to other mothers with more experience and that.”*
*(DGM28, discussion group to mothers)*.

The couple is another pillar on which these women can support themselves to face the changes and to deal with the most demanding aspects of the period. Being in a couple provided support and sustenance through good communication, which facilitates the well-being of women, because they feel more understood and consider it to be the way to improve their relationship.


*“Yes, then, because he has lived… I always find him not as a support, it’s that we are one, that is, we both do the same thing, I don’t consider that he helps me, but we both go and we both do things, maybe he doesn’t clean the bathroom but he comes out of it, what do I know, come on, it’s not: “do I bathe the girl?”, no, he takes and bathes her or… in that aspect it’s not that he helps me, no, it’s that we share the task”.*

*(IM9, in-depth interview with mothers).*



*“We talk apart from the child about our things because it’s a moment of change; I feel that he accompanies me, and I accompany him. For the moment I think we are doing well (laughs)”.*

*(IM4, in-depth interview with mothers).*


This support is, in turn, the main way to deal with the intimate aspects of the couple. These women need to be understood and understood by their partners in the face of the emotions and sensations that arise in sexual encounters after childbirth.


*“The communication with my partner is very good and also helps me a lot in everything, which makes me feel very comfortable and ease my relationships.”*

*(OFM14, online forum to mothers).*


To cope with the new situation, women described how important it is to accept the changes in the postpartum period, which is an advantage over negative emotions that prevent them from motherhood fully experience.


*“For me, having three children. Physically, during my pregnancy, I’ve had good all three pregnancies and postpartum… I looked beautiful all the time. I was happy to be a mother again. And, yes, your body has changed, but yeah! That’s how it is. I’m not the only one, it happens to every woman in the world. Anyway, so it hasn’t influenced me…”*

*(DGM34, discussion group to mothers).*



*“Indeed, you learn to accept yourself”.*

*(DGM33, discussion group to mothers).*


Through facing the changes and being aware of the strength they demonstrate, this empowerment positively influences their self-image.


*“I don’t care because I have something that makes up for it, right? That’s my daughter. And so, I… understand that my body has done wonderful things, and that I have to give it more importance than the pure purely aesthetic aspect.”*

*(IM1, in-depth interview with mothers).*



*“Well, I have felt empowered with the last birth”.*

*(IM12, in-depth interviews with mothers).*


Individual management makes women feel significant and they find time to enjoy and take care of themselves through walks, exercise, or meeting friends, and physical activity to reduce stress are also described as positive strategies in the management of emotions.


*“I have been able to dedicate more time to myself, to the children and to me, so that has made me more relaxed”.*

*(DGM28, discussion group to mothers).*



*“Look for time to take care of yourself and pamper yourself.”*

*(DGM36, discussion group to mothers).*



*“But I believe that also resuming the practice of some physical exercise has been more than anything, for having those two hours that are for me.”*

*(DGM34, discussion group to mothers).*



*“And also, when you are in a bad mood, it lifts you up like crazy, to do sport. Because before I had this one, I started to do it, and not because of the physical aspect, but because I felt better, mentally”.*

*(DGM33, discussion group to mothers).*


### 3.3. Strategies for Coping with Life Changes Affecting Sexuality

The strategies that women have adopted to cope with some of the changes that have altered the way they relate to each other are presented. Women described the search for affection and the desire to care for their partner, as well as having experienced a change in sexual preferences and the need to arrange encounters, as strategies to cope with the new situation that they are experiencing with the arrival of the baby; we obtained 133 fragments in relation to these strategies. The women highlighted their eagerness to take care of their partner, cultivating their relationship through encounters in order to face the lack of intimacy.


*“Very good because for me it’s like that, a moment that I have that right now there are not many such moments, we have a hard time finding the moment, so it’s a moment in which, I don’t know, for me it’s like I share, because of the way he is, I don’t know, I feel good. It brings me closer to him”.*

*(IM9, in-depth interview with mothers).*



*“I am satisfied with being able to have some intimacy with my partner…”*

*(OFM19, online forum with mothers).*


In order to be able to enjoy the encounters, they describe agreeing on the encounters and looking for the right moments to be together and to be able to have moments of intimacy with their partner. They try to combat the lack of availability of time presented by the high demand of the baby and the exhaustion that accompanies it.


*“I mean, it’s a little bit like… it’s forced to look for moments, you know? Although then you enjoy it more than that first time, I was afraid”.*

*(DGM25, discussion group to mothers).*



*“And then we are going to do the strategy of: “come on, we are going to be like teenagers” […] I am going to try, or we want to try to do it this way, so that at least it is not too forced, or something… I don’t know that there is a little bit of desire at least on my part, because not… because not, I don’t feel like it at all”.*

*(IM8, in-depth interview with mothers).*


For women, their own encounters are the basis for affection and care. Demanding a different sexuality, which becomes a moment of intimacy, companionship, closeness, and sensitivity, focused on the skin, touch and caresses for the couple.


*“It’s no longer about having the relationship itself but about sharing a moment of us, you know, so maybe that makes me think more about wanting to have that moment”.*

*(IM9, in-depth interview with mothers).*



*“In my case, sexual desire has been directed more towards affection, the feeling of being understood, supported and accompanied. I don’t have a genitalized sexual desire (neither with myself nor with others), but rather body, skin (caresses, massages). I think it is due to breastfeeding, to the dyad with the baby. I give the baby everything he needs, and sometimes, I would like to be the one who feels cared for and protected”.*

*(OFM13, online forum for mothers).*


Similarly, sexual preferences have been modified, characterizing this stage by its poor versatility in positions, games and practices. They are shorter encounters, with preferences defined by the objective of finishing and avoiding pain.


*“Me, at the time of the day also, because before, I was a morning person. And now, as she sleeps at night, you must do it at night, otherwise in the morning, it’s like I said, a stress of: ‘Come on, come on, the child is crying, and I don’t know what’”.*

*(DGM29, discussion group to mothers).*



*“No, the thing is that maybe they are…I mean, it’s like…. Of course, unconsciously, maybe I want it to be quicker, to go more to the point (laughs)”.*

*(IM8, in-depth interview with mothers).*



*“Now, I enjoy oral sex more than penetration, so to speak”*

*(IM4, in-depth interview with mothers).*


### 3.4. Relationship between Categories of Strategies for Sexual Relations in the Postpartum Period

Strategies become necessary for women from the outset and arise simultaneously with the desire to resume sexual relations. Physical and emotional changes are present from childbirth and together they bring significant changes to cope with.

Figure 1 shows the network of connections between the strategies adopted by women when resuming sexual activity during the postpartum period. This network represents a deeper reflection by the research group and allows readers to explore the relationships between the strategies in order to further understand the phenomenon.

In relation to sexuality, women need to find alternatives focused on how to cope with their sexuality, and they achieve this by agreeing on encounters, and adapting sexual activities. In this way, the encounters themselves become their own strategy to facilitate, care for, and improve the couple’s relationship, as described by the women. In the same way, the improvement of partner relationships contributes to more successful sexual encounters since support and communication are key to feeling understood by their partners.

On the other hand, women show the need for affection that they sometimes obtain through sexual encounters, since they are moments of intimacy between the couple. These encounters influence and are related to women’s self-esteem, sometimes boosting their perceived self-image, and in others, favored by feeling desired by their partner. Having this individual time is essential for women to invest in taking care of themselves and exercising.

Having this time boosts self-esteem as it makes it easier for women to focus on their own needs, particularly those related to physical care. Finally, acceptance of the new situation is a determining factor in improving sexuality. Gaining an understanding the situation would improve the perception of one’s own body and would make it possible to understand that this is a new stage in which sexuality has been modified and, therefore, encounters and preferences have been influenced. This acceptance would also be a determining factor for women to allow themselves space and time to take care of themselves; understanding that, to better manage the new situation, it is essential to take care of themselves.

## 4. Discussion

The present study explored the strategies women use to address the difficulties that emerge from childbirth and up to 6 months postpartum in resumption of sexual activity. In contrast to a recent study [1], which determined these through in-depth interviews with Irish women who were in a late postpartum period (mean 27 months postpartum), the present study focused on the early postpartum period (up to 6 months), studying the phenomenon in the first months postpartum. We learned about the experiences of the restart of sexual encounters without suffering distortions, since, as some studies have shown, memory can be altered over time [43,44]. It is of great interest to researchers to know how women cope with the changes that appear at the stage in which women restart their sexuality after childbirth, because for many of these women it is a time of great change [26]. This period was determined considering that, between 6 and 12 months postpartum, women begin to solve the problems that affect sexuality, improving sexual function, although without reaching pre-pregnancy levels [10,11,12,13]. On the other hand, compared to the study conducted by O’Malley et al., the present study exhibited a greater depth in phenomenon thanks to the triangulation of data collection methods (in-depth interviews, discussion groups, and online forum), and by conducting it in different locations [1].

The fact of choosing a qualitative study from a phenomenological perspective was used to deeply understand the reality of these women in the restart of relationships and subsequent encounters through their feelings, and not only focusing on specialized guides [8]. For this reason, the study focused on the strategies that these women present through their experiences and how they influence the management of their sexuality during the early postpartum period.

Although studies suggest that many women do not resume intercourse in the 6–8-week period after childbirth [5,6,28,45,46] in others, as was the case in the present study (63.88%), most women resumed relationships during this period [11,47]. The cause may be related to the fact that most of the sample in this study did not suffer lacerations in vaginal delivery (27.77%) or they were minor if they did occur (63.88%). According to Jawed-Wessel and Sevick, women who have had vaginal births report the earliest time to resume sexual encounters [11]. One of the biggest problems encountered by women when resumption of sexual activity is the high prevalence of pain in the perineal area, including the fear of feeling it or damaging the area, which has a major effect on their sexual identity, and directly affects intimate relationships with their partners [21]. Many women, worried about the footprint that pain can leave in their relationships, decide to resume coital relations, motivated by maintaining the relationship, putting their partner’s desires before their own [1,48]. As in the study by O’Malley et al., some women in the present study used sexual intercourse with their partner to test the state of their perineum after childbirth and to assess functionality after childbirth [1].

Dyspareunia becomes one of the most recurrent problems in the restart of sexual relations after childbirth [10,18,48], with 40% of the sample experiencing it at 3 months, 28% of the sample experiencing it at 12 months, and 1 in 5 women still experiencing pain at 18 months [18]. Women approach pain from different aspects, such as the use of lubricants, adopting positions that involve less pain, and controlling depth of penetration. These results are consistent with the results from of O’Malley’s study [1]. In other cases, the couple adjusted sexual practices during this period, as also reported by Pardell-Dominguez. Women explained that they needed to control penetration, rhythm, and the possibility of stopping when it was impossible [49]. As other authors have already revealed, practices that included penetration were not the preferred ones in this period [10]. The privileged practices were oral sex and masturbation, they are the first practices to be taken up, as women showed that they felt more confident with them, moving away from the socially constructed concept or sexual expression based on intercourse [3,10,11,21,50]. Indeed, as Bender et al. rightly describes, for many women physical intimacy does not include coital relations, thus defining a new concept of sexuality based on emotional intimacy rather than physical intimacy [7].

Women transition to a new reality from the moment of childbirth which requires changes and coping methods. The lack of time for self-care, the physical change linked to pregnancy and subsequent childbirth, the change of priorities, and the high demand of the baby lead to an increase in stress that is reflected in self-esteem [17]. Authors, such as Serrano, support the contributions provided by our participants, in which they revealed that self-care through sports activity or grooming (make-up, clothing style, etc.) brought positive benefits to their own self-image [21]. Bender et al., in their qualitative study, also reflected the benefits of doing sport and feeling attractive [7]. Other studies indicated that feeling partner desire from their partner produced a positive effect on mother’s self-esteem, which was reflected in an increase in sexual desire and predisposition to have sex with their partner [17,50,51].

This period brings with it a restructuring of the woman and it is common for them to express that they feel disconnected from themselves and their bodies, without being able to control them [21]. Similarly to Bender et al., the current study indicated that body acceptance is fundamental for healthy sexual functionality. Other authors have shown that, after childbirth, many women recognize experiencing changes through childbirth that increases their knowledge and respect for their bodies, appreciating what their bodies have been able to achieve, strengthening a relationship or feeling of empowerment that brings them femininity to their figure and makes them feel more attractive [1,10,19,20,21,22,23,52].

On the other hand, the postpartum period is considered a time of uncertainty, in which the mother takes on a new role associated with motherhood that must be intertwined with other roles that are formed in the family. This transition to motherhood, again, makes women emotionally vulnerable, exposing many insecurities [3,53]. Our participants described that sharing with others who are experiencing the same situation as them, both in motherhood support groups and breastfeeding groups, brought them serenity and emotional well-being. This support has been described in previous studies [20]. In addition, women feel the need for support through the advice of health professionals who can address concerns about baby and postpartum care. This need is supported by many other studies, which also show that women do not feel accompanied during the postpartum period, because, according to them, medical check-ups during this period are insufficient [7,8,14,19,29,54,55].

The postpartum period is intrinsically a transformation of women, couples, and families. These changes, in the case of the participants of this study, manage to deal with these changes with their partner. The couple’s commitment to upbringing or sharing household tasks leads to an increase in emotional satisfaction that is closely linked to the couple’s sexual satisfaction [6,48,56,57]. Other studies endorse these findings, in relation to the fact that the communication in the couple, that the woman feels understood and supported, gives the couple a window of intimacy that facilitates satisfaction [7,10,58,59]. In the same way, they describe how connection and closeness with their partner positively influences their sexuality, due to a new way of understanding and expressing intimacy as a couple [3,10,23,49,60].

Women and their partners are not prepared for possible changes and how they may impact on their intimate relationship [23,48]. This fact demonstrates the need for sex education and interventions focused on addressing the concerns that arise in relation to sexuality postpartum [19,28]. In addition to providing information that fits the reality that couples live to create a broader definition of postpartum sexual health [3,7], couples must be encouraged to explore new sexuality, free of emotional burdens such as guilt, which are based on beliefs or expectations of previous sexuality [3,8,17]. Acceptance of the changes and the new situation can lead women to experience sexuality in a more satisfying ways than experienced previously [3,21,23,48].

### 4.1. Clinical Implications

The results of the current study determined the strategies that women use when resuming sexual relations. For health professionals, knowledge of these strategies allows them to take actions that are more in line with the phenomenon that women experience during the first six months after childbirth regarding their sexuality. In this way, women can be approached from a more holistic perspective.

The conclusions promote the contextualization of sexuality in a period full of changes that must be faced in order to improve couple satisfaction and, therefore, their sexuality.

### 4.2. Strengths and Limitations

As the current study describes the women’s experiences with their sexuality, it is possible that some participants showed some reluctance to speak openly in front of a health professional or in the case of the discussion groups in front of the rest of the mothers, because it may still be considered a taboo issue that they find difficult to talk about [40,61]. Another limitation of the study is that the women in the sample were heterosexual, depriving us of the knowledge of how women with non-heteronormative relationships experience this stage. On the other hand, the fact that the women in the study had a medium–high socio-cultural level prevents learning about strategies in other socio-cultural settings.

## 5. Conclusions

These women, in the first six months after childbirth, face a variety of challenges and show strategies for coping with them. Strategies to facilitate sexual encounters in the context of physical changes are the use of lubricants and changing sexual positions or practices. They face emotional changes through support and good communication from their partner, support from other women, time for oneself, and acceptance of the changes of motherhood. In order to cope with life changes that influence sexuality, women seek to nurture their relationships through affection and sexual encounters, which they adapt to sometimes feel satisfied.

## Figures and Tables

**Figure 1 ijerph-19-00950-f001:**
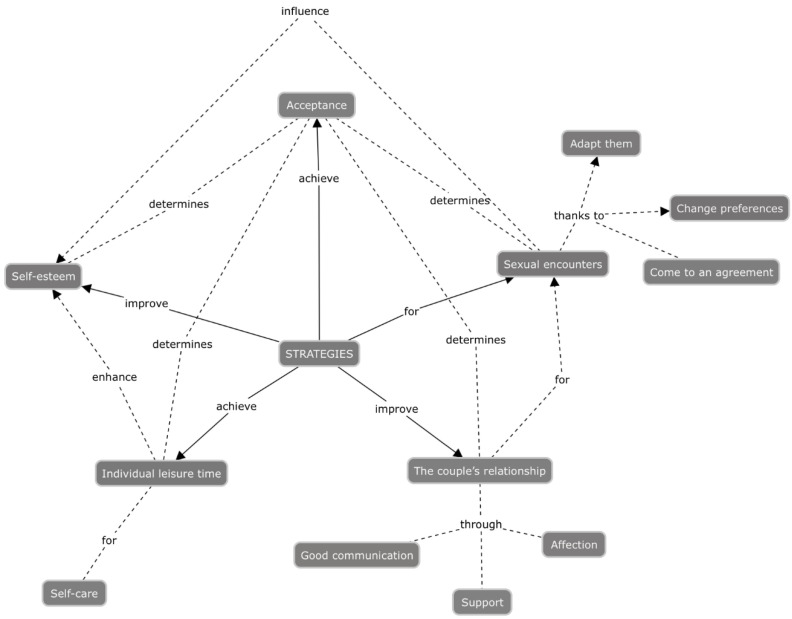
Network of strategies adopted in sexual relations by women in the first 6 months postpartum period.

**Table 1 ijerph-19-00950-t001:** Issues covered in the interviews, the discussion group, and the online forum.

TOPICS
Experience of the postpartum stage.
Feeling when resumption of sexual activity.
Phases of sexuality: sexual desire, arousal, orgasm. Identification of possible influencing factors.
The perception of lubrication.
The state of sexual satisfaction during this period.
Experience of pain during sexual relations.
Identification of concerns, fears associated with pain.
Identification of body image and deepening in relation to the feelings aroused.
Deepening of the family environment.
Deepening of the couple’s relationship, identifying possible changes after the birth of the baby.
Identification of fears and insecurities that may arise after the birth of the baby.
Emotional situation of the woman during this period. Postpartum sadness and/or depression.
Inquire about the birth experience.
Sexual attitudes in the postpartum period.

**Table 2 ijerph-19-00950-t002:** Socio-demographic data of the participants.

Characteristics		n	%
Years			
	Between 25 and 30 years old	2	5.55
	Between 31 and 35 years old	24	66.66
	Between 36 and 39 years old	10	27.77
Parity			
	Primiparous	21	58.33
	Secundiparous	12	33.33
	Multiparous	3	8.33
Marital Status			
	With a partner/single	28	77.77
	Married	8	22.22
Educational level			
	Secondary Education	6	16.66
	Higher Education	26	72.22
	Master’s or Doctorate	4	11.11

Note: n, number of participants; %, percentage of participants.

**Table 3 ijerph-19-00950-t003:** Gynecological and obstetric data, characteristics of the postpartum period, and restart of relations.

Characteristics	n	%
Type of Delivery		
Vaginal without injury	10	27.77
Vaginal episotomia or minor tears	23	63.88
Vaginal with 3rd degree tears	1	2.77
Dystocic birth or delivery	4	11.11
C-section or Caesareans	2	5.55
Breastfeeding		
Exclusive maternal	28	77.77
Mixed	4	11.11
Artificial	4	11.11
Resumption of sexual activity		
Not taken up	3	8.33
6–8 weeks after childbirth	23	63.88
3 months after childbirth	7	19.44
6 months after childbirth	3	8.33
Postpartum period of participation		
Less than a month	0	0
Between 1 and 3 months	8	22.22
Between 3 and 4 months	9	25
Between 5 and 6 months	19	52.77

Note: n, number of participants; %, percentage of participants.

**Table 4 ijerph-19-00950-t004:** Strategy categories and codes.

Categories	Sub-Categories	Codes	Frequencies
Strategies to address physical changes		Adaptations in the encounters	27
		Pain and other dysfunctions	32
Strategies for dealing with emotional changes	Body image	Improve self-esteem	13
		Attraction to the couple	11
		Feeling like a woman	8
	Emotional management	Support from other mothers	7
		Need for a partner	26
		Partner support	21
		Good communication	9
		Acceptance	20
		Feeling powerful	5
		Self-care	9
		More individual time	13
Strategies to address life changes		Taking care of the couple	48
		Arrange encounters	16
		Need for affection	23
		Sexual preferences	46

## Data Availability

Data are held securely by the research team and may be available upon reasonable request and with relevant approvals in place.

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
