# Peer review of "Recovering Sexuality after Childbirth. What Strategies Do Women Adopt? A Qualitative Study"

_ijerph, 2022, doi:10.3390/ijerph19020950_

Round 1
Reviewer 1 Report
I am reviewing “Recovering Sexuality after Childbirth: What Strategies Do Women Adopt? A Qualitative Study” for the International Journal of Environmental Research and Public Health. The study is generally well conceived, run, and described and it makes a solid contribution to the literature. Although I do have any substantive comments for change in the paper, I make suggestions for grammar changes throughout the paper.
The title uses three clauses, when it should say “Determining the Strategies for Recovering
Sexuality after Childbirth in a Qualitative Study”. Line 15 should say “This study aimed to determine the strategies”. Line 17 should say “approach used three data”. Line 22 should say “support, and understanding were the ones that women used to improve”. Line 26 should say “Personal time facilitates”. Line 29 should say “partner, couple time, personal time, and adapting”. Line 30 should say “as the provide insight”. Line 31 should say “the findings will be able to transmit to”. Lines 32-33 should say “relations to improve satisfaction”.
Lines 46-50 should say “Women may tend to delay sexual relations because…changes…they are experiencing…Although most women tend to return to sexual relations between the third and sixth month postpartum, some authors suggest that many women experience at least”. Line 59 say “show that these factors”. Line 60 should say “women showing reduced sexual”. Line 62 should say “satisfaction for the couple”. Line 65 should say “Most studies”. I have no idea what lines 65-68 are trying to say. Please clarify. Line 68 should say “exposed strategies”. Line 70 should say “changes and it showed”. Line 71 should say “described women’s”. Line 73 should say “et al. was performed…perineal trauma described”. Line 75 says “without becoming subject of study” and I have no idea what the authors are trying to say. Line 77 should say “despite this period being the time when women”. Line 78 should say “The current study tried to determine the strategies”. The authors should avoid one-sentence paragraphs. Line 88 should say “study, which included two”. Line 93 should say “arises to deeply evaluate the way many women adapt”. Line 94 should say “6 months and directly affect”.
Line 103 should say “included, and they were not excluded based on who had given”. Line 106 should say “were excluded along with those mothers who”. Line 107 should say “childbirth, experiencing problems”. Line 114 says “these”, but these what? Line 115 should say “guide(Table 1).” Line 116 should say “physiotherapists: two qualitative”. Line 118 should say “were agreed upon by the”. Line 120 should say “home, and in the case”. Line 122 should say “room using the Facebook”. Line 125 should say “Table 1”. Page 128 is blank. Line 137 says “they worked”, but who worked? Line 151 should say “36 Spanish white women”. Line 152 should say “drop out of it.” Line 153 should say “Tables 2 and 3, respectively.” Line 154 should say “sample of these discussion groups”. Line 165 should say “these categories”.
Lines 183-184 should say “physical alterations, such as…4), and the fear”. Line 193 should say
“to improve it.” This paragraph and other paragraphs that are not in quotes should not be one sentence; they should contain at least two sentences. Line 224 says “makes a little get up” and it is confusing. Lines 245-246 should say “Being in a couple provided support and sustenance through good communication, which facilitates”. Line 255 says “meme” but I think the authors mean “me”. Lines 279-282 should say “Individual management makes women feel vital and they find time to enjoy and take care of themselves through walks”. Line 299 should be moved up to line 288. Line 310 should say “partner. They try to combat”. Lines 319-320 should say “sexuality, which becomes a moment….focused on the skin, touch, and caresses for the couple.” Line 344 ends with a preposition “with”. Line 346 should say “when resuming sexual”.
Line 374 should say “explored the strategies”. Line 376 should say “In contrast to a recent study [1] determining these strategies”. Line 378 should say “The present study focused on”. Line 379 should say “We learned about”. Lines 383-384 says “Being for many women…” and I have no idea what the authors are trying to say. Line 388 should say “exhibited a greater”. Line 391 should say “The qualitative format…perspective was used to deeply”. Line 397 should say “Although studies suggest that”. Line 398 should say “48], as was the case”. Line 399 should say “47]. The pause may be related”. Line 401 should say “or they were minor if they did occur…Wessel and Servick”. Line 406 should say “area, which majorly affect”. Line 408 should say “footprint that pain can leave”. Line 409 should say “motivated by couple care, putting”. Line 410 should say “et al., some…study used”. Line 414 should say “with 40% of the sample experiencing…months, 28% of the sample experiencing it at 12 months”. Line 415 should say “Women approach pain”. Line 416 should say “aspects, such”. Line 418 should say “with the results from the O’Malley”. Lines 419-420 should say “Women explained that they needed”.
Lines 427-428 are confusing. Lines 429-430 should say “Women transition to a new reality demanding changes and coping from the moment of childbirth.” Line 433 should say “Authors, such as Serrano,” Line 435 should say “et al., in her qualitative study,”. Line 437 should say “feeling partner desire…on mother’s self-esteem”. Line 442 should say “Like…., the current study indicated that body acceptance is”. Line 445 should say “childbirth that increases their knowledge”. Line 448 should say “them feel attractive”. Line 450 should say “motherhood that must”. Line 452 should say “vulnerable, exposing many”. Line 455 should say “well-being. This support has been described in previous studies.” Line 465 should say “finding, in relation”. Line 469 should say “influences their sexuality”. Line 472 says “This shows” but what shows? Line 474 should say “In addition to providing”. Lines 474-475 should say “live to create…7], couples must be encouraged to explore”. Lines 477-478 should say “Acceptance of the changes and new situations can ….sexuality in more satisfying ways than they experienced previously”. Line 481 should say “The results of the current study determined the strategies that”. Lines 486-487 should be linked with the previous paragraph. Line 489 should say “As the current study describes the”. Line 491 should replace “since” with “because”. Line 500 should say “These women”. Lines 504-505 should say “, which they adapt to sometimes feel satisfied.”
Author Response
I am reviewing “Recovering Sexuality after Childbirth: What Strategies Do Women Adopt? A Qualitative Study” for the International Journal of Environmental Research and Public Health. The study is generally well conceived, run, and described and it makes a solid contribution to the literature. Although I do have any substantive comments for change in the paper, I make suggestions for grammar changes throughout the paper.
The title uses three clauses, when it should say “Determining the Strategies for Recovering Sexuality after Childbirth in a Qualitative Study”. Line 15 should say “This study aimed to determine the strategies”. Line 17 should say “approach used three data”. Line 22 should say “support, and understanding were the ones that women used to improve”. Line 26 should say “Personal time facilitates”. Line 29 should say “partner, couple time, personal time, and adapting”. Line 30 should say “as the provide insight”. Line 31 should say “the findings will be able to transmit to”. Lines 32-33 should say “relations to improve satisfaction”.
Authors: Thanks for your appreciations. The changes have been done.
Lines 46-50 should say “Women may tend to delay sexual relations because…changes…they are experiencing…Although most women tend to return to sexual relations between the third and sixth month postpartum, some authors suggest that many women experience at least”. Line 59 say “show that these factors”. Line 60 should say “women showing reduced sexual”. Line 62 should say “satisfaction for the couple”. Line 65 should say “Most studies”. I have no idea what lines 65-68 are trying to say. Please clarify.
Authors: Thanks for your suggestions. The idea from 65-68 has been rewritten.
Line 68 should say “exposed strategies”. Line 70 should say “changes and it showed”. Line 71 should say “described women’s”. Line 73 should say “et al. was performed…perineal trauma described”. Line 75 says “without becoming subject of study” and I have no idea what the authors are trying to say.
Authors: Thanks for your suggestions. The idea from line has been deleted.
Line 77 should say “despite this period being the time when women”. Line 78 should say “The current study tried to determine the strategies”. The authors should avoid one-sentence paragraphs. Line 88 should say “study, which included two”. Line 93 should say “arises to deeply evaluate the way many women adapt”. Line 94 should say “6 months and directly affect”.
Authors: Thanks for your appreciations. The changes have been done.
Line 103 should say “included, and they were not excluded based on who had given”. Line 106 should say “were excluded along with those mothers who”. Line 107 should say “childbirth, experiencing problems”. Line 114 says “these”, but these what? Line 115 should say “guide(Table 1).” Line 116 should say “physiotherapists: two qualitative”. Line 118 should say “were agreed upon by the”. Line 120 should say “home, and in the case”. Line 122 should say “room using the Facebook”. Line 125 should say “Table 1”. Page 128 is blank. Line 137 says “they worked”, but who worked? Line 151 should say “36 Spanish white women”. Line 152 should say “drop out of it.” Line 153 should say “Tables 2 and 3, respectively.” Line 154 should say “sample of these discussion groups”. Line 165 should say “these categories”.
Authors: Thanks for your appreciations. The changes have been done.
Lines 183-184 should say “physical alterations, such as…4), and the fear”. Line 193 should say“to improve it.” This paragraph and other paragraphs that are not in quotes should not be one sentence; they should contain at least two sentences. Line 224 says “makes a little get up” and it is confusing. Lines 245-246 should say “Being in a couple provided support and sustenance through good communication, which facilitates”. Line 255 says “meme” but I think the authors mean “me”. Lines 279-282 should say “Individual management makes women feel vital and they find time to enjoy and take care of themselves through walks”. Line 299 should be moved up to line 288. Line 310 should say “partner. They try to combat”. Lines 319-320 should say “sexuality, which becomes a moment….focused on the skin, touch, and caresses for the couple.” Line 344 ends with a preposition “with”. Line 346 should say “when resuming sexual”.
Authors: Thanks for your recommendations. The changes have been done.
Line 374 should say “explored the strategies”. Line 376 should say “In contrast to a recent study [1] determining these strategies”. Line 378 should say “The present study focused on”. Line 379 should say “We learned about”. Lines 383-384 says “Being for many women…” and I have no idea what the authors are trying to say.
Authors: Thank you for your consideration. Line has been rewritten.
Line 388 should say “exhibited a greater”. Line 391 should say “The qualitative format…perspective was used to deeply”. Line 397 should say “Although studies suggest that”. Line 398 should say “48], as was the case”. Line 399 should say “47]. The pause may be related”. Line 401 should say “or they were minor if they did occur…Wessel and Servick”. Line 406 should say “area, which majorly affect”. Line 408 should say “footprint that pain can leave”. Line 409 should say “motivated by couple care, putting”. Line 410 should say “et al., some…study used”. Line 414 should say “with 40% of the sample experiencing…months, 28% of the sample experiencing it at 12 months”. Line 415 should say “Women approach pain”. Line 416 should say “aspects, such”. Line 418 should say “with the results from the O’Malley”. Lines 419-420 should say “Women explained that they needed”.
Authors: Thanks for your appreciations. The changes have been done.
Lines 427-428 are confusing.
Authors: We appreciate the proposal clarify the information. This paragraph has been modified following your indications.
Lines 429-430 should say “Women transition to a new reality demanding changes and coping from the moment of childbirth.” Line 433 should say “Authors, such as Serrano,” Line 435 should say “et al., in her qualitative study”. Line 437 should say “feeling partner desire…on mother’s self-esteem”. Line 442 should say “Like…., the current study indicated that body acceptance is”. Line 445 should say “childbirth that increases their knowledge”. Line 448 should say “them feel attractive”. Line 450 should say “motherhood that must”. Line 452 should say “vulnerable, exposing many”. Line 455 should say “well-being. This support has been described in previous studies.” Line 465 should say “finding, in relation”. Line 469 should say “influences their sexuality”. Line 472 says “This shows” but what shows? Line 474 should say “In addition to providing”. Lines 474-475 should say “live to create…7], couples must be encouraged to explore”. Lines 477-478 should say “Acceptance of the changes and new situations can ….sexuality in more satisfying ways than they experienced previously”. Line 481 should say “The results of the current study determined the strategies that”. Lines 486-487 should be linked with the previous paragraph. Line 489 should say “As the current study describes the”. Line 491 should replace “since” with “because”. Line 500 should say “These women”. Lines 504-505 should say “, which they adapt to sometimes feel satisfied.”
Authors: Thanks for your appreciations. The changes have been done.
Reviewer 2 Report
The manuscript explores the strategies women use to address the difficulties that
emerge from childbirth and up to 6 months postpartum in resumption of sexual activity.
The authors analyzed over 36 people. The presented study combines data from several
disciplines, which has an added value. A publication that represents high quality, also taking
into account the qualitative analysis of the results obtained. However, I have a few doubts
about the manuscript in its present form. One will be discussed in detail below to suggest
improvements.
1. Title & Abstract
a. I believe that the title covers the main aspect of the work.
b. I believe that the abstract covers the main aspect of the work.
2. Introduction
a. The introduction correlate to the theme of article.
3. Material and Methods
a. Metohods:
i. the methods are clear and replicable
b. Material:
i. I propose in the subsection participants to present information about
the number of recruited people, and how many people completed the
study
ii. There is no information as to why the full group did not complete the
program. What was the reason for the resignation?
4. Results
a. Results: are relevant and novel. Data plausible.
5. Discussion
a. No issues regarding the Discusson
6. Conclusion
a. The conclusions correlate to the results found
7. Figures & Tables
a. No issues regarding the Figures and Tables
Author Response
The manuscript explores the strategies women use to address the difficulties that emerge from childbirth and up to 6 months postpartum in resumption of sexual activity. The authors analyzed over 36 people. The presented study combines data from several disciplines, which has an added value. A publication that represents high quality, also taking into account the qualitative analysis of the results obtained. However, I have a few doubts about the manuscript in its present form. One will be discussed in detail below to suggest improvements.
Authors: We greatly appreciate your initial comments. They greatly encourage us to continue working in this line.
- Title & Abstract
- I believe that the title covers the main aspect of the work.
- I believe that the abstract covers the main aspect of the work.
- Introduction
- The introduction correlate to the theme of article.
- Material and Methods
- Methods:
- the methods are clear and replicable
- Material:
- I propose in the subsection participants to present information about
the number of recruited people, and how many people completed the
study. There is no information as to why the full group did not complete the
program. What was the reason for the resignation?
Authors: Thanks for your suggestions. The information has been explained in participants section.
- Results
- Results: are relevant and novel. Data plausible.
- Discussion
- No issues regarding the Discusson
- Conclusion
- The conclusions correlate to the results found
- Figures & Tables
- No issues regarding the Figures and Tables
Authors: Thanks for your considerations.
Reviewer 3 Report
Thank you for the invitation.
This manuscript involves some very interesting facts and results from a group of female individuals. I have some recommendations and ideas for this study.
1) From the Introduction Chapter, does the study follow ant theoretical frameworks as the means? If so, may you please tell the readers about the theoretical framework? If not, why no theoretical frameworks were used?
2) May you please edit Page 4 as it is entirely empty?
3) As for the Chapter Two (methodology), may you please add a section about the ethical consideration as Section 2.5? It is important to tell the readers the ways of data protection.
4) In a standardised journal, it is recommended that a paragraph should involve at least four lines. ((Line 183, 184; Line 192-193 etc.)) Please edit the short paragraphs into longer paragraph. This recommendation applies to the entire manuscript as the reviewer cannot point out all problems.
Author Response
Thank you for the invitation. This manuscript involves some very interesting facts and results from a group of female individuals. I have some recommendations and ideas for this study.
1) From the Introduction Chapter, does the study follow ant theoretical frameworks as the means? If so, may you please tell the readers about the theoretical framework? If not, why no theoretical frameworks were used?
Authors: Thanks for your suggestions. The aim of the study was to know women perceptions under a biophychosocial framework.
2) May you please edit Page 4 as it is entirely empty?
Authors: We greatly appreciate your comments. Page 4 has been removed.
3) As for the Chapter Two (methodology), may you please add a section about the ethical consideration as Section 2.5? It is important to tell the readers the ways of data protection.
Authors: Thank you for your consideration, we are sorry we were not clear with this information. We add the information requested after conclusion in a special paragraph.
4) In a standardised journal, it is recommended that a paragraph should involve at least four lines. ((Line 183, 184; Line 192-193 etc.)) Please edit the short paragraphs into longer paragraph. This recommendation applies to the entire manuscript as the reviewer cannot point out all problems.
Authors: We appreciate your proposed modification. Paragraphs have been joined in the text.